# Density fluctuations, homeostasis, and reproduction effects in bacteria

Shahla Nemati[1], Abhyudai Singh [2], Scott D. Dhuey[3], Armando McDonald [4], Daniel M. Weinreich[5] & Andreas. E. Vasdekis [1✉]

Single-cells grow by increasing their biomass and size. Here, we report that while mass and size accumulation rates of single *Escherichia coli* cells are exponential, their density and, thus, the levels of macromolecular crowding fluctuate during growth. As such, the average rates of mass and size accumulation of a single cell are generally not the same, but rather cells differentiate into increasing one rate with respect to the other. This differentiation yields a density homeostasis mechanism that we support mathematically. Further, we observe that density fluctuations can affect the reproduction rates of single cells, suggesting a link between the levels of macromolecular crowding with metabolism and overall population fitness. We detail our experimental approach and the "invisible" microfluidic arrays that enabled increased precision and throughput. Infections and natural communities start from a few cells, thus, emphasizing the significance of density-fluctuations when taking non-genetic variability into consideration.

[1] Department of Physics, University of Idaho, Moscow, ID, USA. [2] Electrical and Computer Engineering, University of Delaware, Newark, DE, USA. [3] Molecular Foundry, Lawrence Berkeley National Laboratory, Berkeley, CA, USA. [4] Department of Forest, Rangeland and Fire Sciences, University of Idaho, Moscow, ID, USA. [5] Department of Ecology and Evolutionary Biology, Brown University, Providence, RI, USA. ✉email: andreasv@uidaho.edu

Across all domains of life, cell growth relies on a series of processes through which cells synthesize new components, replicate their genetic material, increase their size, and eventually divide[1–4]. As such, growth is a key parameter in cellular physiology[5], evolution[6], the production of high-value chemicals[7], as well as human, animal, and plant health[8,9]. Recent investigations at the single-cell level have revealed significant variability in the rates of growth among clonal cells[10]. This form of non-genetic variability has been attributed to fluctuations in enzyme abundance[11], generally emanating from the stochastic nature of gene expression[12–17]. Variability in the reproduction rates between isogenic cells has also been observed[18–20]. In this context, some cells divide considerably sooner or later than the population average, thus, yielding population-level fitness effects that occur at shorter timescales than what mutations can confer[18–21].

Commonly, single-cell growth is investigated by recording the elongation rates (i.e., the cell length, area, or volume per unit time). These, size-based, investigations have unraveled key size homeostasis mechanisms, including the critical accumulation of division proteins and timing of chromosome duplication[22–26]. To a similar end, size-based investigations have informed about the mutation dynamics of single cells and resulting fitness effects[27]. In parallel, single-cell growth has also been examined by recording the dynamics of mass accumulation[28]. Essentially, these measurements capture the underlying metabolic dynamics of nutrient conversion to building blocks, such as amino acids, lipids, and nucleotides[28]. In this context, mass-based investigations have unmasked the exponential nature of mass production[29], as well as the presence of ATP-driven high-frequency mass fluctuations[30]. Moreover, mass-based investigations have revealed that the growth rate of mammalian cells is not constant across the cell cycle[31,32], and the influence of cellular noise on the trade-offs between the naturally evolved and engineered metabolic pathways[33].

Clearly, growing cells need to coordinate both size and mass accumulation, with the latter being enthalpically more pertinent than the former[33]. Cellular size and mass are linked through dry-mass density (dry-density henceforth), namely: the number of molecules per unit volume, or alternatively the level of macromolecular crowding in a microorganism[34,35]. Unlike previous, population-level readouts[36,37], analyses at the single-cell level reveal non-negligible cell-to-cell variability in dry-density, as displayed by way of example in Fig. 1a. Here, a 9% coefficient of variation at mixed growth stages and at birth (by means of synchronization via microfluidic tracking[23]) was observed. Such cell-to-cell dry-density variability suggests that cellular (or molecular) noise effects may be at play[12–17].

Concomitantly, dry-density has also been reported to scale in a species-specific manner[34] with a key role in the folding and stability of key proteins[38]. These observations suggest that a, potentially evolvable, density homeostasis mechanism may also be present. However, and despite the significant discoveries pertaining to cell size regulation[22–25], the regulation and outcomes of the non-genetic variability of dry-density remain less understood. Here, some exceptions pertain to recent reports of dry-density scaling in proportion to the cell's surface-to-volume (S/V) ratio in *E. coli*[39], and the spatiotemporal variation of the dry-density of fission yeast during the cell cycle[40]. Importantly, it is also not known how density variability might influence the metabolic and reproduction rates of single cells[41,42], with the latter being explicitly linked to the overall population fitness[18–21,43,44].

## Results

### Single-cell density measurements

Addressing these knowledge gaps requires assays that can quantify the dynamics of both the density and size of single, growing cells at high-throughput rates.

Quantitative-phase imaging is an ideal candidate to probe these dynamics in a non-invasive manner[32,33,45–48]; in these schemes, however, the dynamic nature of a growing microcolony can yield substantial loss of information. Specifically, interferometric imaging schemes that rely on spatially coherent illumination are limited in spatial bandwidth, which can in turn constrain the homogeneity of the reference field (i.e., the halo effect)[49]. Such inhomogeneities become detrimental when multiple cells reside in close proximity (e.g., when growth is confined to 2D[50]), or cells are imaged in the vicinity of dielectric discontinuities (e.g., microfluidic walls). Light scattering between cells[51], or between cells and dielectric discontinuities[52] can also incur information loss in imaging modalities that rely on spatiotemporally coherent illumination unless dedicated backpropagation algorithms are implemented[53,54].

To overcome these shortcomings, we constructed a microarray that enables dynamic tracking of dry-density and size of multiple single *E. coli* cells with minimal light scattering between cells and between cells and dielectric discontinuities (Fig. 1b). To achieve this, we adopted an 1D immobilization strategy[55,56] that positioned cells at locations that eliminate cell crowding and cell-to-cell scattering. Second, we employed a polymer matrix that became 'invisible' upon contact with water, thus, eliminating scattering between cells and the microfabricated features. Both of these characteristics uniquely enabled the dynamic tracking of single-cell size, mass, and density for up to 6–7 generations (Fig. 1b). In this context, nutrients and stimuli were supplied through vertically integrated membranes or microfluidics (Methods, Supplementary Fig. 1). Further, inspired by Moore's Law in microelectronics, we applied electron-beam lithography to define multiple 1D constrictions at micron-scale distances between them[57]. This lithography step increased the resulting throughput rates (i.e., the number of observations per unit area) by more than one order of magnitude relative to conventional 2D growth approaches (Supplementary Fig. 2).

### Growth differentiation

The combination of the "invisible" 1D microarray with spatial light interferometric imaging (SLIM) enabled the precise quantification of cellular size (approximated by its area, see Methods), dry-density (determined from the measured phase delay through the cell[32], Methods), and dry-mass (through the area product with dry-density[46], Methods). This analysis revealed that while the size and mass of single *E. coli* cells increased exponentially (Fig. 1d, inset and Supplementary Fig. 3), cellular dry-density was not constant during growth (Fig. 1d). Contrary to previous population-level readouts of cellular density[36,37], we observed that dry-density undergoes non-monotonic increases or decreases during growth for most cells (Fig. 1d). We did not observe density fluctuations of similar magnitude in fixed *E. coli* cells (Fig. 1d), suggesting that the observed dynamics in live cells are not due to technical noise. To a similar end, we observed that density fluctuations persist even at higher temporal resolution (Methods), characterized by 'smoother' variations with time than those of Fig. 1d. It is also worth mentioning that density fluctuations have also been observed recently by others in *E. coli*[39] and fission yeast[40] without, however, undergoing further analysis. Interestingly, the overall magnitude of these fluctuations varied between cells, while some cells exhibited overall positive and others negative average density fluctuations during growth (Fig. 1d).

We hypothesized that single-cell density fluctuations during growth could affect the average rates of mass or size accumulation throughout the cell cycle. To assess this hypothesis, we quantified the average accumulation rates of size ($\gamma_{size}$) and mass ($\gamma_{mass}$) during the cell cycle (defined as $A(t) = A_b \cdot e^{\gamma_{size} \cdot t}$ and

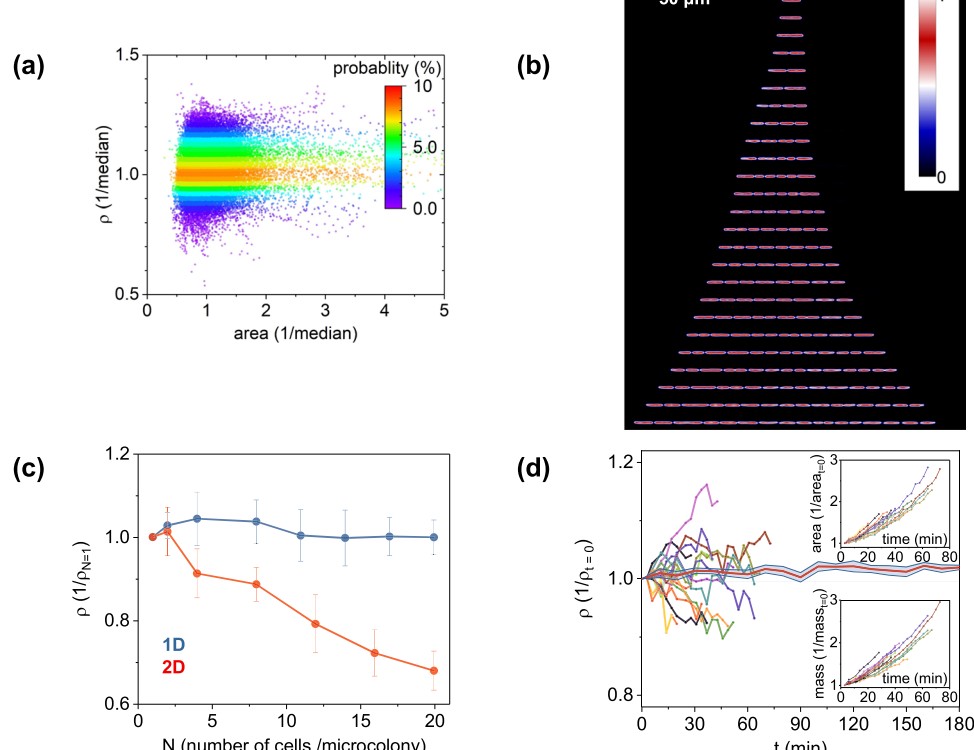

**Fig. 1 Quantitative-mass imaging and "invisible" microfluidics reveal cell-to-cell variability in dry-density and density fluctuations during growth.**
**a** Cell-to-cell dry-density plotted as a function of cell size; graph plots single-cell snapshot data at various stages along the growth cycle ($n = 35,000$ observations). **b** Microcolony expansion from one cell to four generations via quantitative-mass imaging; the vertical direction, with color coding representing cell density (normalized at t = 0, where $N = 1$). **c** Microcolony density (normalized at t = 0, $N = 1$) during expansion using 1D and 2D growth assays; data points and error-bars represent the average and standard deviation of $n = 8$ independent measurements; time-dependent density differences in 2D were statistically significant (one-way ANOVA, $F(6,49) = 51.8$, $p \ll 0.001$); no such evidence was found for the 1D assays (one-way ANOVA, $F(7,56) = 0.8$ and $p = 0.5$). **d** Density, mass, and area growth curves of individual *E. coli* cells from birth to division; all parameters are normalized at t = 0 and color coding represents the size, mass, and density of the same cell (i.e., the curves do not represent the dynamics of multiple cells pooled together); the horizontal red line denotes the dynamics of the normalized density (with respect to t = 0) of 10 fixed *E. coli* cells (DH5α, fixed by overnight incubation in 2% glutaraldehyde, followed by 3× PBS washing) over time (red line denotes the average and blue-shaded area denotes the 95% confidence intervals).

$M(t) = M_b \cdot e^{\gamma_{mass} \cdot t}$, Methods). We observed that these two rates were generally not the same (i.e., $\gamma_{size} \neq \gamma_{mass}$) per cell, but rather clonal cells differentiated into two (continuous) subpopulations: one that exhibits higher rates of size accumulation ($\gamma_{size} > \gamma_{mass}$) and one that reverses this behavior ($\gamma_{size} < \gamma_{mass}$, Fig. 2a). As per our original hypothesis, growth differentiation ($\gamma_{size} \neq \gamma_{mass}$) was essentially found to be driven by the underlying density fluctuations during the cell cycle. Specifically, cells with density fluctuations that were on average positive during the cell cycle differentiated into higher rates of mass accumulation (i.e., $\gamma_{size} < \gamma_{mass}$); conversely, on average negative fluctuations maximized the rates of size accumulation (i.e., $\gamma_{size} > \gamma_{mass}$, Fig. 2b).

In further exploring correlates of growth differentiation, we found that this form of differentiation can be statistically predicted (Supplementary Table 1) from the cellular dry-density at birth (Fig. 2c). Specifically, cells born with lower density than the population median tend to exhibit higher rates of mass accumulation ($\gamma_{size} < \gamma_{mass}$) and vice-versa. Ultimately, density variability at birth can be attributed to the innate randomness or stochasticity, or noise in cellular physiology[17]. This form of stochasticity can include the asymmetric partitioning of biomolecules upon division, as recently shown for single gene products between *E. coli* sisters[58]. Here, we observed a similar form of asymmetry with daughters (at birth) and mothers (at division) exhibiting statistically significant dry-density differences. Further, we observed that daughters were born either at higher or at lower

dry-density than their mothers (Fig. 2d and Supplementary Table 2), with dry-mass differences between daughters to their mothers exhibiting negative correlations (Fig. 2d, inset).

**Density homeostasis.** Concomitantly, we observed that density fluctuations (and the resulting growth differentiation) subsided under the MIC-level pressure from ampicillin (AMP), a bactericidal antibiotic that inhibits cell wall biosynthesis and division (Fig. 3a). Specifically, upon exposure to antibiotics, single *E. coli* cells exhibited variable responses: ~12% of the population either died rapidly or stopped growing, while 88% expressed a filamentous, growing but non-dividing phenotype[59]. The observation of decreased density fluctuations upon AMP exposure pertains specifically to non-dividing filamentous cells and suggests a potential relationship between density fluctuations and cell division.

Intrigued by these findings, we explored whether a density homeostasis mechanism may be at play. To this end, we first linked the median density fluctuations during growth (dρ/dt) with the resulting differentiation behavior and the ratio of birth and division densities ($\rho_{division}/\rho_{birth}$). We observed that cells exhibiting $\gamma_{size} > \gamma_{mass}$ reached division with lower dry-density than their density at birth ($\rho_{division} < \rho_{birth}$, Fig. 2b); conversely cells exhibiting $\gamma_{mass} > \gamma_{size}$ concluded their cycle with higher dry-density at division than at birth ($\rho_{division} > \rho_{birth}$, Fig. 2b). We

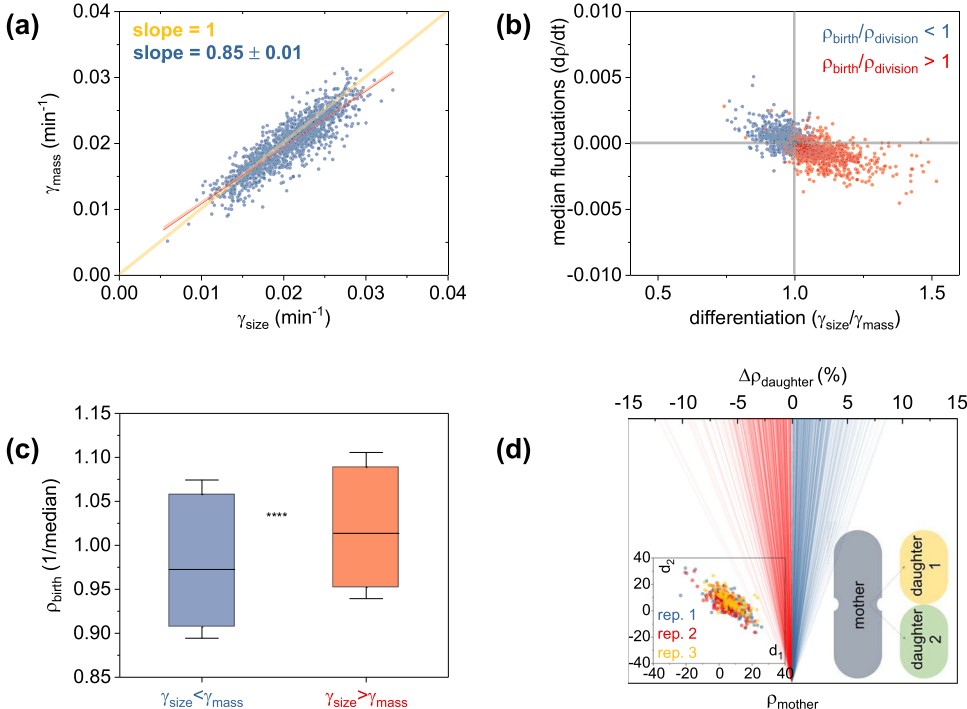

**Fig. 2 Single bacteria do not necessarily exhibit identical growth rates in mass and size, and are prone to asymmetric partitioning of biomolecules upon division. a** Growth differentiation with some cells maximizing area ($\gamma_{size}$) and others biomass ($\gamma_{mass}$) accumulation; graph represents the cumulative response of three replicates, with each replicate presented separately in Supplementary Fig. 4; red line corresponds the linear fit of the experimental data (shaded areas are the 95% confidence intervals) and the yellow line represents a slope of 1. **b** Median density fluctuations during growth ($d\rho/dt$, $y$-axis) as a function of growth differentiation ($\gamma_A/\gamma_M$, $x$-axis); $d\rho/dt$ were calculated as the median value of all density fluctuations (namely: $d\rho = \rho_{i+1} - \rho_i$ in the $dt = t_{i+1} - t_i$ timeframes—see Methods) during the cell cycle; similarly, the growth rates in cell mass and size were calculated by exponential fits throughout the cell cycle, as detailed in the Methods section; color coding corresponds to increased (blue) or decreased (red) density prior to division; graph plots the cumulative response of three replicates, with each replicate presented separately in Supplementary Fig-. 5. **c** Growth differentiation ($\gamma_A/\gamma_M$) dependence on cellular dry-density at birth (normalized over the median); boxcharts represent the 25–75% of the combined three replicates with each replicate plotted separately in Supplementary Fig. 6; whiskers display the 20–80% range and asterisks denotes statistical significance (Mann–Whitney test: $U = 203746$, $p < 0.001$, with additional statistical tests reported in Supplementary Table 1). In support of this finding, we also plot the differentiation dependence on the density at birth in Supplementary Fig. 7a. **d** Division asymmetry in dry-density, as noted by the density differences ($\Delta\rho_{daughter}$ %) between each daughter ($\rho_{daughter-i}$ at birth) to its mother ($\rho_{mother}$ at division), also displayed in inset. Blue (red) traces correspond to density increases (decreases) upon division, and asterisks denote statistical significance of nonzero daughter density differences from their mother (One Sample Wilcolxon Signed Rank Test, $W = 554931$, $Z = 28.11$, $p \ll 0.001$). This graph represents the cumulative response of all biological triplicates, with each replicate presented separately in Supplementary Fig. 8, along with the respective statistical tests in Supplementary Table 2. Inset plots the daughter-daughter correlation of the dry-mass differences ($d_1$ and $d_2$, %) to their mother (color coding represents each replicate).

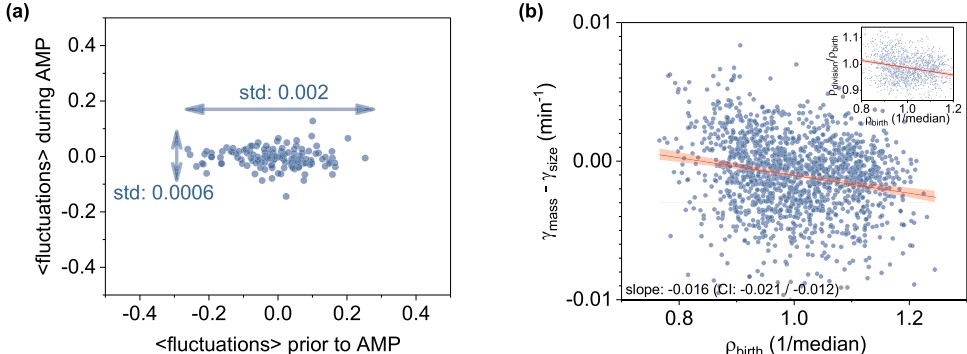

**Fig. 3 Density homeostasis and density fluctuations subsidence upon division inhibition. a** Decrease of density fluctuations under the ampicillin (AMP) pressure; legends denote the standard deviation of fluctuations before (*horizontal arrow*) and during the ampicillin treatment (vertical arrow); graph represents the cumulative response of three replicates with each replicate presented separately in Supplementary Fig. 9. **b** Density homeostasis as evidenced by the monotonic decrease of $\gamma_{mass} - \gamma_{size}$ with respect to the newborn cell density; inset displays the decreasing ratio of cell density prior to division ($\rho_{division}$) over the cell density at birth ($\rho_{birth}$) as a function of dry-density at birth ($\rho_{birth}$). Both the main and inset graphs plot the combined three replicates, with each replicate presented separately in Supplementary Fig. 10 and Supplementary Fig. 11, respectively.

reasoned that this observation is potentially linked to a density homoeostasis mechanism, where density fluctuations maintain cellular density closer to the population average.

To support this mechanism, we considered a simple mathematical model of density fluctuations ($d\rho/dt$) during growth. Based on the notion of exponential size and mass growth (Fig. 1d and Supplementary Fig. 3), then density dynamics can be expressed as $d\rho/dt = (\gamma_{mass} - \gamma_{size}) \times \rho(t)$ (Methods). If $\gamma_{mass}$ and $\gamma_{size}$ do not depend on density then the above model is not homeostatic (even when $\gamma_{mass} = \gamma_{size}$). This is because the slightest noise in $\gamma_{mass}$ or $\gamma_{size}$[11] would enforce density fluctuations to grow unboundedly[60,61]. In contrast, density homeostasis arises by making $\gamma_{size}$ and $\gamma_{area}$ density-dependent. We experimentally verified this dependence, as evidenced by the monotonic decrease of $\gamma_{mass} - \gamma_{size}$ with respect to the newborn cell density (Fig. 3b). The slope of this function was negative with a $-0.016$ slope [$-0.021$, $-0.012$; 95% bootstrap confidence interval (CI), Methods]. The inset on the same plot displays the also decreasing function of the density ratio at division over birth ($\rho_{division}/\rho_{birth}$) with a $-0.33$ slope [$-0.42$, $-0.27$; 95% bootstrap CI] with respect to the newborn density. This decreasing trend also supports density homeostasis, reflecting the control of cellular density in the form of negative feedback[62].

**Single-cell reproduction rates.** Various factors are known to regulate the rates of reproduction or the inverse of the cell cycle duration (i.e., the reciprocal time between two cytokinesis events, $\tau^{-1}$), including cell size at birth, rates of elongation, and timing of chromosome duplication[23,24,26]. To this end, we found that our data also support that both the rates of elongation ($\gamma_{size}$) (Fig. 4a) and cell size at birth (Supplementary Fig. 14) correlate with the rates of reproduction of a single cell. We also notice a non-negligible cell-to-cell variability at high rates of elongation and large birth size, as also observed by others[63]. Indicatively, cells with an equal to or greater than 0.041 min$^{-1}$ rates of elongation (~1.4× above the population average) exhibit 28% coefficient of variation (CV) in reproduction rates (Supplementary Fig. 12). Similarly, cells with an equal to or greater than 4.5 μm$^2$ size/area at birth (~1.4× above the population average) yield a 60% CV in the rates of reproduction. Such levels of cell-to-cell variability suggest that other regulatory layers may act in concert with elongation rates or cell size at birth to modulate the rates of reproduction.

To explore the presence of such additional layers, we investigated how the reproduction rates of single cells may be imparted by the mass at birth ($m_{birth}$), mass accumulation rates

($\gamma_{mass}$), density fluctuations ($d\rho/dt$), and growth differentiation ($\gamma_{size}/\gamma_{mass}$). In this context, we observed that, similar to cell size at birth, mass at birth also correlates with the rates of reproduction, albeit at comparable levels of cell-to-cell variability as cell birth size (Supplementary Fig. 14). Further, we noted that increased reproduction rates occurred for higher $\gamma_{mass}$ and $\gamma_{size}$ (Fig. 4a). In this context, $\gamma_{size}$ displayed a moderately stronger effect (evidenced by the higher slope in the relationship of $\gamma_{size}$ with reproduction rates, rather than $\gamma_{mass}$ (Supplementary Fig. 12); this effect, however, was not found to be statistically significant due to the overlap of the confidence intervals of these two rates between biological triplicates. Unexpectedly, we observed instead that increased levels of growth differentiation ($\gamma_{size}/\gamma_{mass}$) could also predict high rates of reproduction (Fig. 4b). In this context, increased density fluctuations (either positive or negative on average during the cell cycle) imposed higher rates of reproduction even at lower rates of mass ($\gamma_{mass}$) and size ($\gamma_{size}$) accumulation (Fig. 4b). Similarly, individuals characterized by low rates of reproduction were substantially less abundant at increased levels of density fluctuations (Fig. 4b). This observation indicates that elevated rates of reproduction can be expressed not only by individuals exhibiting elevated rates of mass and size accumulation, or large size and mass at birth but also by those that undergo increased density fluctuations. We note that these results pertain primarily to the conditions that were explored in this work, namely: nutrient-rich and steady-state (i.e., time-invariant) microenvironments.

## Discussion

In summary, we report that while individual *E. coli* cells accumulate size and mass at exponential rates (Fig. 1d and Supplementary Fig. 3), their dry-density fluctuates non-monotonically during growth. Also observed by others[39,40], these density fluctuations enable further insight into the origins of the non-genetic density variability between cells (Fig. 1a) and suggest that the average rates of size and mass accumulation during the cell cycle may differ in one cell. In this context, we observed that depending on whether density fluctuations are, on average, positive (or negative) during the cell cycle, clonal subpopulations emerge that exhibit higher average rates of mass accumulation than size (and vice-versa). These phenomena subsided under division inhibition using bactericidal antibiotics, suggesting a potential link between density fluctuations with cell wall biosynthesis and division.

We linked this form of growth differentiation to a density homeostasis mechanism, namely the tendency of cells with density that differs from the population average to accordingly adjust

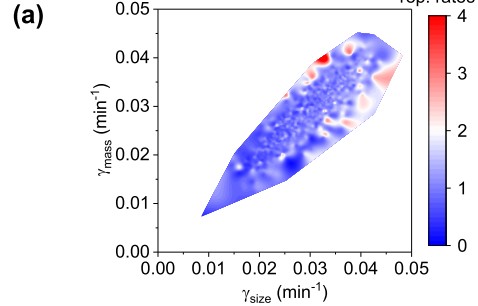
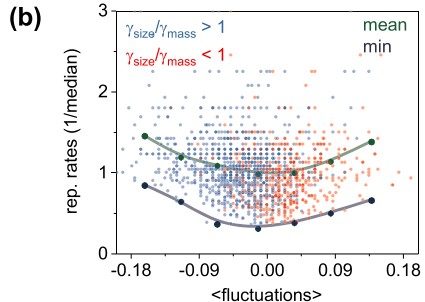

**Fig. 4 Density fluctuations impact replication rates of single bacteria. a** 3D representation of growth differentiation ($\gamma_A - \gamma_M$ relationship), with each single-cell observation color coded by its replication rate level; graph represents the cumulative response of three replicates with each replicate presented separately in Supplementary Fig. 13. **b** Single-cell replication rates plotted as a function of density fluctuations; blue and red data points represent single-cell observations (color coded by their level of differentiation); green and purple points represent the averaged binned data and minimum replication rates levels at different levels of fluctuations; similarly, graph plots the combined three replicates, with each replicate presented separately in Supplementary Fig. 15.

their density. Specifically, cells exhibiting higher rates of mass production reach division at a higher density state (Fig. 2c). Conversely, cells born at a density that is higher than the population's, exhibit higher rates of size accumulation, thus, decreasing their density upon division. Following the first report of this work[64], a density homeostasis mechanism was also evidenced in fission yeast, with 11 out of 76 total observations born at higher dry-density exhibiting negative density changes during their cycle[40]. To a similar end, density homeostasis or "homeocrowding" mechanism that echoes the one reported here was also put forward earlier by van de Berg and colleagues[34]. Here, the authors posited that cells can maintain optimal dry-density given the established correlates between cell size and protein mass with nutrient availability[65].

We supported this density homeostasis mechanism with an analytical model. This model suggests the density dependence of the rates of mass and size accumulation, primarily through proteome allocation towards size increase (see Methods). We also demonstrate that not only the size but also the mass of single *E. coli* cells at the selected growth conditions is consistent with the adder phenotype[22,24,25] (Supplementary Figs. 16 and 17); however, the observed density fluctuations suggest the likely presence of an additional control layer. Our results are also consistent with the recently reported linearity between the ratio of cell surface area (S) and mass (M) with cell length[39] (Supplementary Fig. 18a); we note, however, that the rapid growth regime explored here yields a (mitotic) constriction and, thus, an inflection in the S/V rate towards the end of the cell cycle (Supplementary Fig. 18b) as also reported elsewhere[22].

We also explored density correlates with the rates of reproduction of single cells. Specifically, we found that individual cells that exhibit increased reproduction rates can be identified not only by their size/mass at birth or their rates of size/mass accumulation but also by the amplitude of their density fluctuations (Fig. 4). In this context, we observed that individual cells exhibiting high (low) reproduction rates were substantially more (less) abundant at increased density fluctuations levels (Fig. 4b). Mechanistically, this can be thought of as gratuitous overexpression of growth-required components at birth (e.g., ribosomes)[5], for individuals born with high dry-density (and undergoing overall negative density fluctuations during growth). This notion is in agreement with recent reports of yeast supergrowth, following inhibition of size expansion but not of protein synthesis[66]. Conversely, cells born at low dry-density (and undergoing positive density fluctuations during growth) may permit increased translational degrees of freedom for enzymes and metabolites, and reduced conformational entropic penalties[34]. Both of these effects have been previously shown to accelerate metabolic reactions[67,68].

The explicit link between overall population fitness to the reproduction rates of single cells[18,19,21] suggests that the density fluctuations we observe here may also play a role in the fitness of a population. More explicitly, the Malthusian fitness of a population, defined by its rate of growth (i.e., dN/dt, with N being the number of individuals in the population)[44,69], reduces the rate of reproduction ($\tau^{-1}$) for $N = 1$[70]. Reducing Malthusian fitness to the single-cell level (i.e., $N = 1$) has been attempted before[27], approximated through the rate of elongation ($\gamma_{size}$) of single cells; however, our results suggest that this approximation may not encompass all aspects of reproduction timing of single cells.

Finally, we detail our experimental approach that enabled the dynamic tracking of cellular dry-density with enhanced precision and throughput rates, which is challenging, if not impossible, with conventional microfluidic systems. For enhanced precision, we fused quantitative-mass imaging with 1D microarrays that not only eliminated cell crowding but also became invisible upon contact with water. This approach minimized light scattering at cell-to-cell and cell-to-microfluidics interfaces, thus, preserving key optical information during microcolony expansion (Fig. 1b). We also employed advanced microfabrication to improve the underlying throughput rates by more than one order of magnitude in comparison to existing assays. These methodological approaches can be directly translated to other species to further explore the role of cellular dry-density and related effects in growth, division, and reproduction rates in single cells.

Given that all inoculants and infections start from a single or very few growing cells, we anticipate that the paradigm of density fluctuations, homeostasis, and reproduction rate effects will improve systems and evolutionary biology investigations that take the segregated notion of non-genetic variability into consideration[71].

## Methods

**Strains**. Two strains were used in the reported investigations, namely: *Escherichia coli* DH5α (WT) and the ampicillin-resistant *E. coli* E212K mutant, also derived from DH5α. This derivative was chosen to include one more strain in our density fluctuations observations (i.e., in the absence of antibiotic pressure), and specifically carries the *g628a* mutation (using Ambler numbering[72]) on the *TEM-1* gene on *pBR322*.

**Growth conditions**. WT and resistant strains were grown using a bath incubator (C76, New Brunswick Scientific) at 37 °C and 180 rpm. As a growth medium, we employed the Mueller Hinton broth (Difco 275730, BD). The strains were first passed from agar plates (stored at 4 °C) to 5 ml medium (round bottom polystyrene tubes, VWR) until the early stationary phase, then diluted in 20 ml fresh medium (125 ml glass flasks, Corning) at a 0.01 optical density ($OD_{600}$, $\lambda = 600$ nm, V-1200 spectrometer, VWR), and incubated for 12 h (37 °C, 180 rpm). Overnight cultures were diluted to an $OD_{600}$ of 0.01 (in 20 ml fresh medium), regrown to mid-exponential phase (~3 h), and sampled to perform all reported single-cell experiments. We employed the same procedure to determine the MIC levels of the E212K mutant, as detailed below. All single-cell experiments were performed in triplicates by repeating the abovementioned procedure on different days.

**Minimum inhibitory concentration (MIC)**. We measured the MIC levels of the E212K strain using the microdilution method[73]. Following growth (see above), cultures were diluted to $10^6$ cfu/ml (~0.002 $OD_{600}$) to a volume of 1 ml and transferred to 1 ml of ampicillin (VWR0339, VWR) solution in Mueller Hinton at concentrations ranging from 8196 to 0.0156 μg/ml at a 1.4× step size[74], including a 0 μg/ml control. The suspensions were incubated (37 °C, 180 rpm) for 20 h to determine the MIC level, namely the lowest ampicillin concentration yielding zero OD[73]. We found this level to be at 176 μg/ml (Supplementary Fig. 19), and applied this value in all subsequent microfluidic experiments. The measurement was repeated (three times with mid-exponential phase cultures and four times using stationary phase cultures) yielding the same result.

**Single-cell assays**. Single *E. coli* cells were laterally confined using 1D microarrays and vertically confined via a top-integrated membrane (Supplementary Fig. 1), enabling size, density, and mass tracking of single cells for up to 6–7 generations. The 1D microarrays were fabricated by electron-beam lithography in SU8 (using our previously reported procedure[75] and further detailed in the following section), subsequently transferred to PDMS, and then to a UV curable polymer that was index-matched to water (Bio-133, My Polymers). The total thickness of the polymer film after imprinting was ~0.5 mm, thus, accommodating the working distance of all employed imaging objectives. This thickness was achieved by depositing ~100 μL of the liquid prepolymer on the stamp. Nutrients were provided by a doped membrane or a microfluidic channel. The latter was applied in the reported antibiotic experiments to yield dynamic switching between medium and ampicillin conditions. Below we detail the microfabrication procedure of the 1D microarrays, including the approach we followed in nutrient supply.

**1D microarray fabrication**. The 1D microarrays were first realized in the SU8 photoresist (2002, Microchem) on silicon (Si) wafers using electron-beam lithography (VB300, Vistec). We employed SU8 for its ultra-high sensitivity, enabling the definition of submicron features over large areas at high speeds[75]. Following plasma cleaning (5 min), dehydration (180 °C for 5 min) of the Si wafer, and spin-coating (5 mins, 5000 rpm) with SU8, the resulting films were soft-baked at 65 °C (1 min) and at 95 °C (1 min). Following exposure at a 2 μC/cm² dose, the films were baked at 65 °C (2 min) and then at 95 °C (2 min), followed by development in propylene glycol methyl ether acetate (Sigma–Aldrich). The exposed patterns consisted of 1.2 μm wide and 1 μm deep lines, spaced at 1 μm distances. The total length and width of the patterns were 6 × 0.5 mm². Following the development, the microarrays were transferred from SU8 to PDMS (Sylgard 184, Dow Corning) by cast-molding lithography at a 10:1 monomer-to-catalyst ratio[76,77]. Finally, the PDMS pattern was transferred to a UV curable polymer (Bio-133) to generate

microarrays with the pattern originally displayed on the Si wafer (i.e., 1.2 µm wide and 1 µm deep trenches spaced at 1 µm distances). For this step, the PDMS slab was placed in a glass petri dish, coated with ~100 µL of Bio-133, and degassed for 2 h. Subsequently, the polymer-coated PDMS stamp was covered with a coverslip and degassed for another 2 h, followed by UV curing (Q4000 Optical Mask Aligner, Quintel). All steps were performed in a cleanroom environment under yellow light. The microarrays were used repeatedly after washing in ethanol (~60 sec) and ultrasonicating in water.

**Nutrient/antibiotics supply**. In the single-cell growth experiments, nutrients were delivered via vertically integrated, nutrient-doped, agarose membranes. We employed a similar strategy for 2D growth in the throughput and information-retention comparisons presented in Supplementary Fig. 2 and Fig. 1c. For 1D growth, 0.5 µl of cells were deposited on the microarrays and immediately covered with the agarose membrane (Supplementary Fig. 1). For 2D growth, cells were first deposited on the nutrient-containing agarose and, after a 5 min drying time, covered using a coverslip. The agarose membranes were prepared by dissolving 1.5% agarose (UltraPure, Invitrogen) in Mueller Hinton broth at 80 °C for 1 h. Approximately 300 µl of liquid agarose was deposited on a coverslip (25 × 50 mm², L × W) and a second coverslip of the same dimensions was immediately mounted on top. The agarose membrane was let dry for 20 min at room temperature, yielding a thickness of ~90 µm. Our second method for nutrient/antibiotics delivery applied in the ampicillin experiments employed microfluidics assembled using five distinct parts that were held together mechanically[78,79] (Supplementary Fig. 1): (1) a PDMS stamp (dimensions: 35 × 20 × 4 mm³, L × W × H) prepared by optical and cast-molding lithography containing the microchannel (dimensions: 11 × 0.5 × 0.1 mm³, L × W × H); (2) the 1D microarray to immobilize cells; (3) a hybrid membrane (detailed below) to confine cells in the vertical direction; (4) two thin aluminum plates that hold parts (1), (2), (3), and the cell-loaded 1D micro-arrays together using four symmetrically placed screws; and (5) a syringe pump (Harvard Apparatus) connected to the microfluidics via Tygon tubing (1548XL, IDEX) and 21 gauge needle tips, set at 20 µl/min. The hybrid membranes (i.e., part 3) that were combined with the microfluidics, were formed by combining dialysis membranes with agarose gels. Specifically, single-layer dialysis membranes (10 µm thickness, cellulose, MWCO: 12–14,000, Fisher) were cut to 20 × 40 mm² areas. The membranes were boiled in 2% sodium carbonate for 30 min and then transferred to a boiling Tris Hydrochloride solution for another 30 min[80]. Following a water rinse, the membranes were transferred to Mueller Hinton broth and stored at 4 °C for up to 2 weeks. To prepare the hybrid membranes, a dialysis membrane was first placed on a coverslip, covered with 200 µl of liquid agarose, and then with another coverslip. The assembly was left at room temperature for 20 min until the agarose solidified to a ~60 µm thickness. The hybrid membrane confined cells and transported nutrients from the vertically integrated microfluidic channel. The latter was connected with a three-way switch valve (IDEX) and two syringes (one containing the medium broth and the other the ampicillin solution). The medium was first flown through the device for ~50 min, followed by switching to ampicillin and propidium iodide (1 µM)[81] for up to 3 h. In all experiments, the devices were transferred to a temperature-controlled (37 °C) incubator integrated with an inverted microscope.

**Imaging**. Regarding cell growth, we performed quantitative-mass imaging using a spatial light interference microscopy (SLIM) system (Cell Vista Pro, Phi Optics) integrated with an inverted microscope (DMi8, Leica) equipped with an automated stage. In our SLIM system, the quantitative-phase images are formed by projecting the back focal of the imaging, phase-contrast, objective onto a liquid crystal spatial light modulator (SLM). The SLM exhibits 'ring-shaped' phase masks that shift the optical phase of the light wavefront scattered by the sample relative to the unscattered light, as detailed in the original report of this technique[45]. In this way, images representing the relative phase delay of *E. coli* cells (scattered wavefront) with respect to the background (unscattered wavefront) are formed. 3D z-stack images (0.3 µm step size) were acquired using a ×63 (NA 0.7, PH2) or with a ×40 objective (NA 0.6, PH2, at 0.5 µm step size) and a 3.65 µm pixel CCD camera (GS3-U3-28S4M, Point Grey Research). To correct for halo effects, and in addition to arranging the microcolonies in 1D as detailed earlier and presented in Fig. 1c and Supplementary Fig. 2, we also processed the quantitative-phase images to remove any residual halo using the computational procedure described elsewhere[45]. This step increased the background uniformity, thus, enabling a better definition of the cell contour, a key parameter in cell segmentation[82]. A comparison of the quantitative-phase image of a single cell with and without halo correction is displayed in Supplementary Fig. 20. Various locations were imaged every ~3 min (every ~5 min for the ampicillin experiments) using automated routines (Metamorph, Molecular Devices). All single-cell growth experiments were performed in triplicates yielding a total of $n = 1520$ observations of single dividing cells, with each replicate consisting of 442 (*rep. 1*), 573 (*rep. 2*), and 505 observations (*rep. 3*). These observations (and all related subsequent analyses) exclude the very first mother cell. Specific to the dry-density comparison between mothers and daughters (Fig. 2d and Supplementary Fig. 8), density differences were calculated by considering dividing mothers, and both dividing and non-dividing daughters, thus, enabling the consideration of all diving mothers and yielding 412 (*rep. 1*), 582 (*rep. 2*), and 452 (*rep. 3*) observations. Finally, in the ampicillin (AMP) experiments we

did not consider cells with fewer than 4 temporal observations (both prior to and during AMP pressure), as well as non-growing cells (e.g., persisters), yielding 60 (*rep. 1*), 45 (*rep. 2*), and 35 (*rep. 3*) observations.

In the reported throughput analysis, throughput denotes the number of observations per field of view, or alternatively the maximum possible number of microcolonies in a single image. We compared the 1D and 2D assay throughputs at ×20 (NA 0.4, PH1) and ×40 (NA 0.6, PH2) magnification, respectively using quantitative-phase imaging and a 6.5 µm pixel size sCMOS camera (ORCA-Flash 4, Hamamatsu). For 1D, we imaged ~1000 fixed DH5α cells (overnight fixation in 2.5% glutaraldehyde at 4 °C, followed by 3× washing in medium, and diluted at a varying optical density from 0.01 to 0.175), which we introduced in the 1D microarrays. We followed a similar approach in 2D, albeit using live cells that we allowed to grow to microcolonies containing ~20 cells. This was performed in triplicates, with each replicate containing 40 microcolony observations.

**Image analysis**. All quantitative-mass images were processed using ImageJ and Fiji (National Institutes of Health), Metamorph, and MATLAB (Mathworks), as follows: (1) choice of the best-focus plane ($p_i$) and a maximum projection of $p_i$, $p_{i-1}$, and $p_{i+1}$; (2) filtering by median and gaussian blur (ImageJ), 2D deconvolution (*No Neighbors*, Metamorph), and 1D Fast Fourier Transform (FFT, ImageJ); and (3) thresholding via the Maximum Entropy algorithm (ImageJ). All resulting binary images were subsequently subjected to watershed, visual inspection, and—if necessary—manual curation. Following processing, the binary and original quantitative-phase images were assembled into two separate time-lapse stacks, divided into microcolonies, and analyzed with a lineage mapper (ImageJ) to extract lineage trees and track single cells from birth to division.

We selected the abovementioned image processing pipeline for its robustness and reduced computational requirements, as we have previously demonstrated for bacteria and yeast[33,82]. Further, while density fluctuations have been reported by others[39] and we also observe them in high temporal resolution readings (with smoother traces, Supplementary Fig. 21), we performed additional steps to ensure that our observations are not due to cell segmentation or plane selection errors. Specific to cell segmentation, we ensured the validity of our approach by performing the following two analyses. First, we compared the abovementioned with the Otsu and Moments thresholding algorithms. Second, we compared our approach to a 1D segmentation approach that is independent of conventional thresholding algorithms and, thus, possible errors in area segmentation. In this context, we determined the beginning and end of a 1D cell contour (of a constant 6-pixel width) at 20% above the noise floor. All comparisons (Supplementary Fig. 21) yielded moderate differences in the single-cell density dynamics, which upon normalization (at t = 0 or the time of birth) exhibited very high agreement in single-cell density dynamics characterized by greater than 99% Pearson correlation coefficients ($p < 0.001$). This finding suggests that the observed density fluctuations represent a physiological response, largely independent of potential errors during cell segmentation.

Further, we ensured that we selected the proper plane of focus by inspecting the quantitative-phase images of all cells at all collected z-planes, as well as the z-dependence of their phase signal. To this end, we employed a custom MATLAB code that simultaneously displayed cell images of all planes and selected the plane ($p_i$) that exhibited the sharpest image[83]. Following maximum projection between $p_{i+1}$ and $p_{i-1}$, we visually inspected all images to ensure appropriate plane selection. Furthermore, we estimated the induced density uncertainty after intentionally selecting the wrong plane of focus. To this end, we intentionally selected ±1 plane away from focus and computed the resulting single-cell density error (standard error). In this way, we computed a 0.78% uncertainty in the density determination of a single cell due to an experimental error in the plane selection, which is lower than the observed density fluctuations, as displayed in Supplementary Fig. 21.

**Throughput analysis**. We employed ImageJ to quantify 1D and 2D throughput using the previously detailed procedures. To quantify 1D throughput, we used the resulting statistics of 1000 cells to determine the average cell length and the distance of each cell to its nearest neighbor. To quantify 2D throughput, we analyzed images of microcolonies containing up to 16 cells to determine the largest microcolony dimension using the Feret's diameter (ImageJ). In this context, we did not approximate the microcolony as a circle, given that 2D confined *E. coli* microcolonies form dynamic nematic patterns of variable asymmetries and orientations[84], as also displayed in Supplementary Fig. 2b. To compare information loss in 1D and 2D, we analyzed images of 1D and 2D microcolonies containing 20 single cells. Subsequently, we averaged the dry-density of all cells in the microcolony and performed related statistical tests, as reported in Fig. 1c.

**Data analysis**. The following metrics were extracted from each image: area, density, and mass per time-point per cell. Growth rates were computed using these functions of $A(t) = A_b \cdot e^{\gamma_{size} \cdot t}$ and $M(t) = M_b \cdot e^{\gamma_{mass} \cdot t}$ in MATLAB throughout the cell cycle. Density ($\rho$) fluctuations were determined as the median of $d\rho/dt$, where $d\rho$ represents the $\rho(t_{i+1}) - \rho(t_i)$ difference in the $dt = t_{i+1} - t_i$ window. To quantify

the density of single-cells from the measured optical phase delay ($\Delta\Phi$), we used the following expression[32]:

$$\rho = \frac{\lambda}{2 \cdot \pi \cdot \frac{dn}{dc}} \cdot <\Delta\Phi> \tag{1}$$

with $\frac{dn}{dc} = 2 \cdot 10^{-4} \frac{m^3}{kg}$ representing the protein-specific refractive index increment[28], $\lambda$ the wavelength of illumination (centered at 550 nm), and $<\Delta\Phi>$ the experimentally determined phase delay difference between the cell cytosol and the extracellular medium, integrated across the cytosolic area A. We note that Eq. 1 computes the density of a single cell without prior knowledge of the cell area, and applies 2D maximum projection of 3D data (planes $P_{i-1}$ to $P_{i+1}$, where i is the best-focus plane—see Image Analysis subsection). Finally, to determine cell mass, we multiplied cell density (Eq. 1) with the cell area. This has been previously demonstrated in cell mass measurements, where cell thickness cannot be determined as accurately as its area, given the lower axial than the planar resolution of most optical imaging systems[46].

In regards to the throughput analysis, we quantified 1D throughput by performing a nearest neighbor analysis (MATLAB, *knnsearch*, *euclidean*). Specifically, we set the minimum distance to the nearest neighbor equal to the average cell length multiplied by 16 (the expected number of progeny in a microcolony for the duration of our experiments). In this context, we only considered individual cells exhibiting horizontal distances (i.e., in an axis parallel to growth) from the nearest neighbor that were greater than this threshold. We followed a similar procedure to quantify 2D throughput. Here, we performed a 2D nearest neighbor analysis (MATLAB, *knnsearch*, *euclidean*) after determining the largest microcolony size through the Feret's diameter, and using this diameter as a threshold.

**Density homeostasis model**. Let mass $M$ of a single cell grow exponentially with rate $\gamma_M$ during the cell cycle:

$$\frac{dM}{dt} = \gamma_{mass} \cdot M(t) \tag{2}$$

Similarly, area $A$ of a single cell grows exponentially with rate $\gamma_A$:

$$\frac{dA}{dt} = \gamma_{size} \cdot A(t) \tag{3}$$

Then, density (defined as the mass over area ratio) evolves as:

$$\frac{d\rho}{dt} = (\gamma_{mass} - \gamma_{size}) \cdot \rho(t) \tag{4}$$

If $\gamma_{mass}$ and $\gamma_{size}$ are density-independent then the above model is not homeostatic even when $\gamma_{mass} = \gamma_{size}$, as the slightest noise in these rates makes density fluctuations grow unboundedly over time[60,61]. In contrast, density homeostasis arises by making $\gamma_{mass}$ and $\gamma_{size}$ density-dependent. Let

$$e^{\gamma_{mass} - \gamma_{size}} \tag{5}$$

be a monotonically decreasing function $f$ of newborn cell density $\rho_i$:

$$e^{\gamma_{mass} - \gamma_{size}} = f(\rho_i) \tag{6}$$

As such, a newborn with a low density will invest more in mass growth *vs.* area growth. Substituting (6) in (4), the density at the end of the cell cycle is:

$$\rho_f = f(\rho_i)^T \cdot \rho_i \tag{7}$$

where $T$ is the length of the cell cycle. With this, one can write the following iterative model for the newborn densities $\rho_i$ in the $n$th generation:

$$\rho_{i,n+1} = f(\rho_{i,n})^T \cdot \rho_{i,n} + \varepsilon_n \tag{8}$$

Where $\varepsilon_n$ is the noise-induced at division from random partitioning of area and mass. The above model has a unique fixed point given by the solution to the equation:

$$1 = f(\rho) \tag{9}$$

which will be a stable homeostatic set point for $\rho$ in the presence of noise as long as the specialization function $f$ is a decreasing function of density. Equation 6 can be rewritten as:

$$\frac{\gamma_{mass}}{\gamma_{size}} = \gamma_{size} + \log(\rho_i) \tag{10}$$

Equation 10 implies that the $\gamma_{mass}/\gamma_{size}$ ratio within a cell cycle should also be a decreasing function of the newborn cell density.

Using some of the ideas put forward by others[39], we further explored potential mechanisms that could underly density homeostasis. One such mechanism includes:

$$\frac{dA}{dt} = \alpha \cdot M \tag{11}$$

This equation denotes that part of the proteome is dedicated to the increase of cell size, which is congruent with density homeostasis. In this context, Eq. 1 leads to

the following expression for the temporal evolution of density:

$$\frac{d\rho}{dt} = (\gamma_{mass} - \alpha \cdot \rho) \cdot \rho \tag{12}$$

In this case, the density at a steady state is given by:

$$\bar\rho = \frac{\gamma_{mass}}{\alpha} \tag{13}$$

Solving differential Eq. (2), the ratio of densities at the start and end of the cell cycle is

$$\frac{\rho_f}{\rho_i} = \frac{e^{\bar\rho T\alpha}\bar\rho}{(e^{\bar\rho T\alpha} - 1)\rho_i + \bar\rho} \tag{14}$$

where $T$ represents the duration of the cell cycle (or the inverse of reproduction rates). This ratio is a decreasing function of $\rho_i$, as noted in the inset of Fig. 3b, which suggests density homeostasis.

Given that both mass and area increase exponentially as per

$$\frac{dM}{dt} = \gamma_{mass} \cdot M(t), \tag{15}$$

and

$$\frac{dA}{dt} = \gamma_{size} \cdot A(t) \tag{16}$$

Then the model described by Eq. (11) corresponds to $\gamma_{size}$ being an increasing function of density. Indeed, this is the behavior we observe in our experimental data, as evidenced in Supplementary Fig. 7.

An alternative model of the density of homeostasis, also put forward by others[39], is:

$$\frac{dA}{dt} = \alpha \cdot \frac{dM}{dt} \tag{17}$$

This expression corresponds to the ratio $\gamma_{size}/\gamma_{mass}$ being proportional to the density at birth. While we do see a modest increase in $\gamma_{size}/\gamma_{mass}$ as a function of density at birth, we observe a stronger increase with $\gamma_{size}$ as a function of density (Supplementary Fig. 7). These differences suggest that the model presented in Eq. 11 is more likely to be the dominant driver of density homeostasis.

**Adder, sizer, and timer model**. To compare cell size at division ($A_d$) with the adder, sizer, and timer models from the size at birth ($A_b$), we employed the following expression[24]:

$$A_d = 2 \cdot \alpha \cdot \Delta + 2 \cdot (1 - \alpha) \cdot A_b \tag{18}$$

We adapted Eq. (11) to similarly represent cell biomass at division ($M_d$) as:

$$M_d = 2 \cdot \alpha \cdot \Delta + 2 \cdot (1 - \alpha) \cdot M_b \tag{19}$$

In both equations, $\alpha$ varies as: $\alpha = \frac{1}{2}$ for adder, $\alpha = 1$ for sizer, and $\alpha = 0$ for timer. $\Delta$ is the median area ($A_b$) and mass ($M_b$) at birth. The results of these three models are plotted in Supplementary Fig. 16 (for size) and Supplementary Fig. 17 (for mass) and compared to the experimental raw data (scatter plot), a linear regression (based on the experimental data), and the binned experimental data.

**Statistics**. The robust coefficient of variation was derived in MATLAB using the *mad(X,1)* function to first calculate median absolute deviations that we divide with the population's median. ANOVA tests were performed in MATLAB using the *anova1* function. The 95% confidence intervals of all linear regressions were computed in MATLAB by bootstrapping using the *bootci* function ($n = 1000$ samples). Binning was performed in MATLAB using the *Sturges* method and the *histcounts* function. Mann–Whitney, Kolmogorov–Smirnov, two-sample $t$-tests, and all plotted linear regressions were performed in Origin Pro.

**Reporting summary**. Further information on research design is available in the Nature Research Reporting Summary linked to this article.

# Data availability

Key data generated or analyzed during this study are included in this article (and its Supplementary Information), as well as Supplementary Data 1; all data are available from the corresponding author upon reasonable request.

# Code availability

The custom Matlab code developed for the best-focus plane from 3D quantitative-phase imaging stacks can be accessed online through Zenodo[83].

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

## Acknowledgements

A.E.V. and A.M.D. acknowledge support from the U.S. Department of Energy, Office of Science, Office of Biological, and Environmental Research (DE-SC0019249). S.N., D.W., and A.E.V. acknowledge support from the National Science Foundation (OIA-1736253). A.S. acknowledges support from the National Institutes of Health (5R01GM124446). Discussions with James Bull and Larry Forney are also gratefully acknowledged.

## Author contributions

S.N. performed the imaging experiments, image and data analysis, and contributed to the manuscript preparation; A.S. developed the mathematical model of homeostasis and contributed to the manuscript preparation; S.D.D. and A.M.D. contributed to the microfabrication of the microfluidic assays; D.W. contributed the strains and manuscript preparation; A.E.V. acquired funding, overviewed this research, and wrote the manuscript.

## Competing interests

The authors declare no competing interests.
