## [Peer Review File · Communications Biology]

Reviewers' comments:

Reviewer #1 (Remarks to the Author):

The paper deals with the timely problem of density homeostasis in growing cells. Unfortunately, I was not convinced of the conclusions based on the current data presented in the paper.

Main points

- Many of the claims in the paper are not substantiated by a proper quantitative analysis or the use of statistics. For example, the support of exponential growth of mass and area - which is central to this paper - is not appropriately done, but rather hinted at by showing a few curves of mass/area vs. time, which also appear to be the results for pooling together several cells rather than a single-cell?

As another example, around line 153 the authors discuss a "similar form of asymmetry" which is also not clearly evident in Fig. 2d they refer to: what is the signature for the asymmetry? Can one distinguish a real asymmetry from random fluctuations in the histogram presented? There are other examples of claims made and not supported by a quantitative analysis and backed up by statistics.

Similarly, the "fitness" later used in the paper is not properly motivated and in light of numerous recent works related single-cell growth to population dynamics it is far from clear that it is a correct notion of fitness.

- There are several others works recently published on the topic, notably by Sven van Teeffelen's group. It seems that the data presented here is contradictory to the previously published results, yet there is little discussion regarding why this discrepancy exists. A detailed, quantitative comparison would be important, or alternative methods for validating the method utilized in the current paper.

- The presented theoretical model seems quite generic and it is unclear to me how it supports the findings of this work. Sentences such as "We validated this mechanism with a simple mathematical model" simply don't make sense logically - how can a biological mechanism be theoretically validated?

- What are the effects of measurement noise? For example, when finding the effects vanish for filamentous cells, could this be simply due to less (relative) measurement noise for these larger cells?

Minor:

- There is an important typo in the equation on line 175.

Reviewer #2 (Remarks to the Author):

Review of a manuscript COMMSBIO-21-1254 „Density Fluctuations Yield Distinct Growth and Fitness Effects in Single Bacteria.“

The manuscript submitted by Nemati et al. presents a discovery of a phenomenon of fluctuating density of cells during the growth of E. coli bacteria. Authors came to such a conclusion by combining microscopy analysis of a cell size (area) with improved interferometric measurement of cell dry mass. Confining of bacteria in 1D microfluidic channels in the device made of low-refractive index polymer enabled for reduction of light scattering and higher precision of dry mass measurement in comparison to analysis of 2D growth of bacterial microcolonies. Overall, the research presented in the manuscript comprised both observation of a novel biological phenomenon and development of new methodology that can be useful for a broad research community studying dynamics of microbial growth. The article is well-written, and the data is presented in a clear and methodical manner. The results are also supported with meticulous statistical analysis.

I recommend that the manuscript can be published after minor revision.

Minor comments:

1. Authors used polystyrene beads for the normalization of the cell density measurement (Fig. 1d). I suppose that these particles are not optimum objects to be used for this normalization, because both chemical composition and the shape of beads differ significantly from these parameters of bacteria. Ideally, live but non-dividing E.coli bacteria should be used for normalization. In that case, the neutral buffers such as phosphate-buffered saline (PBS) should be used instead of growth media. I think that authors should perform such a simple normalization experiment and add this data to the manuscript.

2. Some technical details are missing in the description (in the SI) of the microfluidic experiments. Authors forgot to mention what is the thickness of biopolymer microarray used for microfluidic 1D cultivation of bacteria, and how this thickness was determined during the casting of this microarray. Also, the diameters of the microfluidic inlet and outlet should be provided. Finally, the description of how the layers of the microfluidic system were held together mechanically is not clear, i.e. whether clamps were used or what was the way to keep these parts together.

Reviewer #3 (Remarks to the Author):

I consider the manuscript by Nemati et al. to be of high quality and suitable for publication in *Communication in Biology*. The presentation and the format of the manuscript is appealing, while the scientific content is of high quality. The experimental and image/data analysis methods are smart and precisely described. The findings and conclusions are well supported by the data. What I particularly find interesting the results on growth differentiation and linking density fluctuations to fitness.

Reviewer #1:

The paper deals with the timely problem of density homeostasis in growing cells. Unfortunately, I was not convinced of the conclusions based on the current data presented in the paper.

General Response: We thank the Reviewer for the detailed feedback and highlighting the timeliness of this work. We considered all points that were raised, and accordingly reorganized the manuscript. Below are our responses to the comments, including the corresponding changes in the revised manuscript **highlighted in green**:

Main points:

[R1-1] *Many of the claims in the paper are not substantiated by a proper quantitative analysis or the use of statistics. For example, the support of exponential growth of mass and area - which is central to this paper - is not appropriately done, but rather hinted at by showing a few curves of mass/area vs. time, which also appear to be the results for pooling together several cells rather than a single-cell?*

Response: While we believe that most findings presented in the original manuscript were sufficiently substantiated (as also noted by Reviewers 2 and 3), we agree that some, parallel, claims could use further statistical analysis. As such, to address the first comment on the exponential nature of mass/size accumulation, we performed the following two revisions:

Revision 1, caption of Fig. 1d:

We further clarify that the term “*same cell*” used in the caption of **Fig. 1d** in the original manuscript pertains to the curves of individual cells, rather than multiple cells pooled together.

From (caption of Fig. 1d, line 569): “*Density, mass, and area growth curves of individual E. coli cells from birth to division; all parameters are normalized at $t = 0$ and color-coding represents the dynamics in size, mass, and density of the same cell...*”

To (caption of **Fig. 1d**, line 569): “Density, mass, and area growth curves of individual *E. coli* cells from birth to division; all parameters are normalized at $t = 0$ and color-coding represents the dynamics in size, mass, and density of the same cell (i.e., these curves represent the dynamics of individual cells and not several cells pooled together) ...”

Revision 2, Sup. Fig 3:

We compared the residuals of the exponential and linear fits of the mass and size accumulation rates, as recently reported by others (in **Fig. 2G**). As displayed in new **Sup. Fig. 3** (referenced on page 6, line 120 in the revised manuscript), smaller residuals in the exponential fits support the notion that area and mass grow more exponentially than linearly:

(a) Typical examples of the exponential nature of size and mass accumulation; data points represent experimental observations while the solid lines represent the linear (red) and exponential (blue) fits. **(b)** Plot of regular residuals for the curves presented in (a). **(c)** Residual sum comparison for all area and dry-mass fits presented in **Fig. 1d**; here, we specifically plot the sum of the absolute residuals for mass and area accumulation during the cell cycle (i.e., $\sum_{birth}^{division} |residual|$).

[R1-2] *As another example, around line 153 the authors discuss a "similar form of asymmetry" which is also not clearly evident in Fig. 2d they refer to: what is the signature for the asymmetry? Can one distinguish a real asymmetry from random fluctuations in the histogram presented?*

Response: The original manuscript included a histogram (**Fig. 2d**) of the mother to daughter dry-density ratio. In this graph, any deviations from unity represent an asymmetry in division (or fluctuations/deviations in the symmetry of the division process). As clarified in the original manuscript, this ratiometric approach has also been adopted by others in the context of the asymmetric partitioning of a single gene product between daughter cells (see *REF 51* in the original and *REF 56* in the revised manuscript).

To ensure, however, that the notion of “*density asymmetry upon division*” is clear, we revised **Fig. 2d** to display the density differences between each daughter cell from its mother. To this end, any deviation from the origin (i.e., $\Delta\rho \neq 0$) represents the asymmetry between the mother density at division and the daughters’ density at birth. We further support this finding with a relevant statistical test. These revisions are detailed below:

Revision 1: Main text

*From (page 7, line 149): “This form of stochasticity can potentially include the asymmetric partitioning of biomolecules upon division. Such asymmetric partitioning was recently evidenced in the division of single gene products between daughter *E. coli* cells⁵¹. A similar form of asymmetry is also evident in our work **Fig. 2d**.”*

*To (page 8, line 150): “This form of stochasticity can potentially include the asymmetric partitioning of biomolecules upon division. Such asymmetric partitioning was recently evidenced in the division of single gene products between daughter *E. coli* cells⁵⁴. **Here, we observed a similar form of asymmetry with daughters (at birth) and mothers (at division) exhibiting statistically significant dry-density differences. Further, we observed that daughters were born either at higher or at lower dry-density than their mothers (Fig. 2d and Sup. Table 2).**”*

Revision 2: Fig. 2d

(d) Division asymmetry in dry-density, as noted by the density differences ($\Delta\rho_{daughter} \%$) between each daughter ($\rho_{daughter-i}$ at birth) to its mother (ρ_{mother} at division), as displayed in the cartoon (inset). Blue (red) traces correspond to density increases (decreases) upon division, and asterisks denote statistical significance (One Sample Wilcoxon Signed Rank Test, $W = 554931$, $Z = 28.11$, $p = 0$) of nonzero daughter density differences from their mother. This graph represents the cumulative response of all biological triplicates, with each replicate presented separately in **Sup. Fig. 8**, along with the individual statistical tests in **Sup. Table 2**.

Revision 3: Fig. S8

Dry-density asymmetry upon division, noted by the density differences ($\Delta\rho_{daughter} \%$) between each daughter ($\rho_{daughter-i}$ at birth) to its mother (ρ_{mother} at division). Blue (red)

*traces correspond to density increases (decreases) upon division and asterisks denote statistical significance (One Sample Wilcoxon Signed Rank Test, **Sup. Table 2**) of nonzero daughter density differences from their mother. The graphs represent the response of each biological triplicate separately.*

[R1-3] There are other examples of claims made and not supported by a quantitative analysis and backed up by statistics.

Response: Without additional information, we could not identify any other claims in the original manuscript that could use additional support. As also noted by Reviewers 2 and 3, the original manuscript supports its main findings (i.e., density homeostasis, fitness effects, and growth differentiation), through the following:

1. Investigation of three independent biological replicates (a common approach in population-level investigations but not in those performed at the single-cell level).
2. Statistical tests for all biological triplicates, including three different statistical tests for density homeostasis (**Sup. Table 1**).
3. Graphs of all biological replicates pooled together (*main manuscript*), as well as each biological replicate plotted separately (*supplementary information*) along with their individual statistical test.

Further, the following statistical analyses were also included in the original manuscript:

4. All linear fits presented in the manuscript included bootstrapped slopes and confidence intervals, as detailed in the *Methods* section. This approach pertains specifically to:
 - i. the differentiation dependence on the density at birth (**Sup. Fig. 7a**);
 - ii. the elongation rate dependence on the density at birth (**Sup. Fig. 7b**);

- iii. the ratio of dry-density at division over birth as a function of the density at birth (**Sup. Fig. 11**);
 - iv. the growth rate by size and mass as a function of single-cell fitness (**Sup. Fig. 12**);
 - v. adder/sizer/timer models for both area (**Sup. Fig. 16**) and mass (**Sup. Fig. 17**).
5. The ampicillin MIC levels measurement using six replicates, with three pertaining to stationary and four to exponential phase cells (**Sup. Fig. 19**).
 6. The investigation of eight replicates and their ANOVA analysis to evidence the precision advantages conferred by our proposed 1D invisible microfluidic method in comparison to 2D, conventional, approaches (**Fig. 1d**).

[R1-4] Similarly, the "fitness" later used in the paper is not properly motivated and in light of numerous recent works related single-cell growth to population dynamics it is far from clear that it is a correct notion of fitness.

Response: In the original manuscript, we attempted to clarify that the term fitness used throughout pertains to the replication rate of one cell and not of the population. In light of this comment and the comment made by Reviewer 3, we revised manuscript to clarify that the focus is on the replication rates of single-cells. We also explain in the Discussion section how/why these reproduction rates reflect the Malthusian fitness at the single-cell level and acknowledge the assistance of Dr. J. J. Bull to this end:

Revision 1: Main text

From (page 9, line 184): "Fitness Effects: Finally, we explored how density fluctuations and the resulting growth differentiation may confound γ_A and γ_M as predictors of single-cell fitness, as previously postulated^{16, 23} or expected from population-level Malthusian models^{16, 37}. Here, we quantify the fitness of a single-cell through the rate of (asexual)

production of progeny, or alternatively the inverse of the life-cycle duration, as also posited by others⁵⁶. We note that this definition pertains to the fitness levels of a single-cell rather than that of the population⁵⁷. Further, this definition coincides with the time required by a cell to complete its cycle as typically reported in size control and homeostasis investigations²¹. Similarly, this definition captures the notion of the instantaneous reproduction capability of single-cells and the rate of gene contribution to the next generation in a constant environment. By experimentally delineating fitness, γ_A , and γ_M , we observed that fitness generally increased with γ_M and γ_A (**Fig. 4a**), with γ_A displaying a moderately stronger effect (evidenced by the higher slope in the relationship of $\gamma_A - \text{fitness}$ than $\gamma_M - \text{fitness}$, **Sup. Fig. 12**); this effect, however, was not statistically significant due to the overlapping confidence intervals and, importantly, the several low fitness individuals persisting at high rates of size and mass accumulation (**Fig. 4a**)."

To (page 9, line 191): "Single-Cell Reproduction Rates: Various factors are known to regulate the rates of reproduction or the inverse of the cell-cycle duration (i.e., the reciprocal time between two cytokinesis events, τ^{-1}), including cell size at birth, rates of elongation, and timing of chromosome duplication^{21, 22, 24}. To this end, we found that our data also support that both the rates of elongation (γ_{size}) (**Fig 4a**) and cell size at birth (**Sup. Fig 14**) correlate with the rates of reproduction of a single cell. We also notice a considerable cell-to-cell variability at high rates of elongation and large birth size, as also observed by others⁶¹. Indicatively, cells with an equal to or greater than 0.041 min^{-1} rates of elongation ($\sim 1.4x$ above the population average) yield 28% coefficient of variation (CV) in rates of reproduction (**Sup. Fig 12**). Similarly, cells with an equal to or greater than $4.5 \mu\text{m}^2$ size at birth ($\sim 1.4x$ above the population average) yield a 60% CV in the rates of reproduction. Such variability levels suggest that other regulatory layers may act in concert with elongation rates or the cell size at birth to modulate the rates of reproduction.

To explore the presence of such additional layers, we investigated how single-cell reproduction rates may be imparted by the mass at birth (m_{birth}), mass accumulation rates (γ_{mass}), and density fluctuations (dp/dt) and growth differentiation ($\gamma_{\text{size}}/\gamma_{\text{mass}}$). In this context, we observed that, similar to cell size at birth, mass at birth also correlates with the rates of reproduction with comparable levels of cell-to-cell variability (**Sup. Fig. 14**).

Further, we noted that increased reproduction rates occurred for higher γ_{mass} and γ_{size} (Fig. 4a)."

Revision 2: Main text

From (page 11, line 235): "Importantly, we report a previously masked link between growth and fitness at the single-cell level that does not conform with previous single-cell postulates²³ and population-level (e.g., Malthusian) models³⁷. Specifically, we found that individual cells that exhibit high levels of fitness can be better identified by their density-fluctuations amplitude than the respective rates of size or mass accumulation (Fig. 4a). In this context, we observed that high (low) fitness individuals were substantially more (less) abundant at increased fluctuations levels (Fig. 4b). Mechanistically, this can be thought of as gratuitous overexpression of growth-required molecular components at birth, specifically for individuals undergoing increased negative density fluctuations; conversely, overall positive density fluctuations of increased levels may enable cells to invest more in mass production than size accumulation, thereby reducing the time required to reach division."

To (page 13, line 266): "Mechanistically, this can be thought of as gratuitous overexpression of growth-required components at birth (e.g., ribosomes)⁵, for individuals born at high dry-density (and undergoing overall negative density fluctuations during growth). This notion is in agreement with recent reports of yeast supergrowth, following inhibition of size expansion but not of protein synthesis⁶⁴. Conversely, cells born at low dry-density (and undergoing positive density fluctuations during growth) may permit increased translational degrees of freedom for enzymes and metabolites, and reduced conformational entropic penalties³². Both of these effects have been previously shown to accelerate metabolic reactions^{65, 66}.

Further, given the emerging relationship between reproduction rates of single-cells to the overall population fitness^{17, 18}, we note that the aforementioned notion of single-cell reproduction rates coincides with the Malthusian fitness equivalent of a single-cell⁴². Our rationale behind this adaptation is that since Malthusian fitness is valid for a population, it should also be valid for its individual members. Overall, Malthusian fitness for a

population is defined by its rate of growth (i.e., dN/dt , with N denoting the number of individuals of the population)⁴². As such, the Malthusian fitness of a single-cell reduces to dt^{-1} , which denotes the inverse of the cycle duration or the reproduction rate of a single-cell, as also posited elsewhere⁶⁷. To distinguish the fitness of a single-cell from that of a population, we propose the term “instantaneous fitness effect”, with instantaneity denoting a moment or instant in phenotypic-space⁶⁸. ”

[R1-5] There are several others works recently published on the topic, notably by Sven van Teeffelen's group. It seems that the data presented here is contradictory to the previously published results, yet there is little discussion regarding why this discrepancy exists. A detailed, quantitative comparison would be important, or alternative methods for validating the method utilized in the current paper.

Response: The Reviewer refers to the recent article from the van Teeffelen group. We would also like to note an earlier work also from the Cheng group, and clarify that both works were published in peer-reviewed journals after this manuscript was submitted to this journal and bioRxiv.

Both published articles focus primarily on the relationship between cellular morphology and dry-density during the cell cycle. In contrast, the work presented in this manuscript addresses the notion of density homeostasis, a timely topic, as noted by all Reviewers, that requires high throughput investigations. Further, our observations corroborate the findings in both articles, and support a previous hypothesis by van de Berg and colleagues that posited that density homeostasis (or “*homecrowding*”) can emerge through the interplay between size and mass growth rates.

In detail:

- As also mentioned in the original manuscript, density fluctuations were observed in the two abovementioned articles without, however, undergoing further analysis. To this end, please refer to **Fig. 2D** in the van Teeffelen group article and **Sup. Fig. 2A** in the Chang group article.

- Metanalysis of the data presented in our original manuscript evidence that the surface to mass (S/M) ratio is also linear with cell length, and exhibits a comparable SEM to the findings reported in the van Teeffelen group article for *E. coli*:

- Density homeostasis is suggested in the article from the Cheng group in a different model system. In this context, ~11 out of ~76 observations (**Fig. 1D**) display negative density variations during the cell cycle, as also detailed in our manuscript.

In terms of differences, we note one with respect to the van Teeffelen group article. In this work, the authors reported that dry-density exhibits a decreasing trend from the moment of birth until a little earlier than the middle of the cell cycle (**Fig. 2F**). The authors attribute this behavior as a response to the similarly decreasing surface to volume (S/V) ratio. Subsequently, dry-density increases until the cell divides. As discussed in the same article, this form of dry-density rate inflection coincides with the time of septum (i.e., Ftsz ring) formation. In support of this claim, the authors present 49 observations.

Our results display a similar decreasing/increasing density rate inflection, albeit only for a subset of observations ($n = 40$). In contrast, when all our observations ($n = 1,520$) are pooled together in the same graph, we observe that the average dry-density at division is comparable to that at birth, without undergoing a considerable decrease at the middle of the cell cycle. These results are shown below:

To explain this difference, we offer two potential reasons:

1. *Growth conditions:* We adopted growth conditions that yield 20-25 min doubling times and considerable cell-to-cell variability. In contrast, the growth conditions presented in the van Teeffelen group article yield a longer cell cycle duration with no (apparent) cell-to-cell variability. This could potentially also explain why S/V rate changes occur much later than the middle of the cell cycle in our system. This is displayed by way of example below for two daughters that are born at the same time but divide (white arrows display the formation of the mitotic constriction) and is in agreement with earlier and more recent reports by others:

2. *Number of observations:* as also noted by Reviewers 2 and 3, the high throughput approach presented in our manuscript ~30-fold more observations and replicates than those reported in the above articles. We cannot comment on the importance of the number of observations on morphological analyses, since this is not focus of this work; however, we can say that homeostasis investigations require a high number of observations. For example, should our analysis be performed with only

the abovementioned 40 observations, then the pooled results would indicate that dry-density is not homeostatic, but rather decreases with each division event.

To clarify these points and integrate all three abovementioned references in the revised manuscript, we performed the following revisions:

Revision 1: Main text

From (page 4, line 68): “Cellular size and mass are linked through dry mass density (dry-density henceforth), namely: the number of molecules per unit volume, a metric that represents the level of crowding within the cytosolic environment.”

To (page 4, line 67): “Cellular size and mass are linked through dry mass density (dry-density henceforth), namely: the number of molecules per unit volume, or alternatively the level of molecular crowding in a microorganism^{32, 33}.”

Revision 2: Main text

From (page 4, line 71): “Unlike previous, population-level, readouts^{32, 33}, single-cell analyses reveal considerable cell-to-cell variability in dry-density, as displayed in **Fig. 1a** with a coefficient variation of 9%; however, and despite the significant discoveries pertaining to cell size regulation¹⁹⁻²², the non-genetic variability in density and its influence on the size-mass coordination during growth remain significantly less understood. A notable exception here is a recent report that cellular density varies with the cell’s surface-to-volume ratio, highlighting the importance of biomass growth in cell size control³⁴.”

To (page 4, line 69): “Unlike previous, population-level readouts^{34, 35}, single-cell analyses reveal cell-to-cell variability in dry-density, as displayed by way of example in **Fig. 1a**. Here, a 9% coefficient of variation at both mixed growth-stages and at birth (by means of synchronization via microfluidic tracking²¹) was observed. Such cell-to-cell phenotypic variability in dry-density suggests that cellular noise effects may be at play³⁶. Concomitantly, dry-density has also been reported to scale in a species-specific

manner³², suggesting that a, potentially evolvable, density homeostasis mechanism may also be present. However, and despite the significant discoveries pertaining to cell size regulation²⁰⁻²³, the regulation and outcomes of the non-genetic variability of dry-density remains less understood. To this end, some exceptions pertain to recent reports of dry-density scaling in proportion to the cell's surface-to-volume (S/V) ratio in *E. coli*³⁷, and the spatio-temporal variation the dry-density of fission yeast during the cell cycle³⁸.”

Revision 3: Main text

From (page 6, line 128): “It is also worth mentioning that density fluctuations have also been reported recently by others in *E. coli*³⁴ without, however, any further analyses on these dynamic phenomena.”

To (page 6, line 127): “It is also worth mentioning that density fluctuations have also been reported recently by others in *E. coli*³⁷ and fission yeast³⁸ without, however, further analysis.”

Revision 4: Main text

From (page 11, line 225): “It is also worth mentioning that density fluctuations have also been reported recently by others in *E. coli*³⁴ without, however, any further analyses on these dynamic phenomena.”

To (page 12, line 241): “Conversely, cells born at a density that is higher than the population's, exhibit higher rates of size accumulation, thus, decreasing their density upon division. Following the first report of this work⁶², a density homeostasis mechanism was also evidenced in fission yeast, with 11 out of 76 total observations born at higher dry-density exhibiting negative density changes during their cycle³⁸. To a similar end, a density homeostasis or “homeocrowding” mechanism that echoes the one reported here was also put forward earlier by van de Berg and colleagues³². Here, the authors proposed that it is possible for cells to maintain optimal dry-density given the established correlates between cell size and protein mass with nutrient availability⁶³.”

Revision 5: Main text

From (page 11, line 231): “It is worth noting, however, that the adder model does not take the observed density fluctuations into consideration, suggesting the presence of an additional control layer that is not represented by the adder phenotype.”

To (page 12, line 252): “We also demonstrate that not only size but also the mass of single *E. coli* cells at the selected growth conditions is consistent with the adder phenotype^{20, 22, 23} (**Sup. Fig. 16 and 17**); however, the observed density fluctuations suggest the likely presence of an additional control layer. Our results are also consistent with the recently reported linearity between the ratio of cell surface area (*S*) and mass (*M*) with cell length³⁷ (**Sup. Fig. 18a**); we do note, however, that the rapid growth regime explored in this work yields a (mitotic) constriction and, thus, an inflection in the *S/V* rate towards the end of the cell cycle (**Sup. Fig. 18b**), as also reported elsewhere²⁰.”

[R1-6] *The presented theoretical model seems quite generic and it is unclear to me how it supports the findings of this work. Sentences such as "We validated this mechanism with a simple mathematical model" simply don't make sense logically - how can a biological mechanism be theoretically validated?*

Response: We attribute this confusion to the selected nomenclature in the original manuscript. As such, we replaced ‘validation’ with ‘support’ throughout the revised manuscript.

[R1-7] *What are the effects of measurement noise? For example, when finding the effects vanish for filamentous cells, could this be simply due to less (relative) measurement noise for these larger cells?*

Response: Similar to the comment by Reviewer 2, we revised the technical noise quantification presented in **Fig. 1d** and present density fluctuation data using fixed cells. Overall, there is technical noise, as with most empirical measurements, but we find that this noise is less than the observed fluctuations. We also offer below a detailed view of the density fluctuations as a function of cell length under antibiotic pressure. As it can be observed, density fluctuations remain at comparable levels, irrespective of the cell length.

Minor:

[R1-8] *There is an important typo in the equation on line 175.*

Response: We have accordingly modified this equation to indicate:

$$\frac{d\rho}{dt} = (\gamma_{mass} - \gamma_{size}) \cdot \rho(t)$$

Reviewer #2

The manuscript submitted by Nemati et al. presents a discovery of a phenomenon of fluctuating density of cells during the growth of E. coli bacteria. Authors came to such a conclusion by combining microscopy analysis of a cell size (area) with improved interferometric measurement of cell dry mass. Confining of bacteria in 1D microfluidic channels in the device made of low-refractive index polymer enabled for reduction of light scattering and higher precision of dry mass measurement in comparison to analysis of 2D growth of bacterial microcolonies. Overall, the research presented in the manuscript comprised both observation of a novel biological phenomenon and development of new methodology that can be useful for a broad research community studying dynamics of microbial growth. The article is well-written, and the data is presented in a clear and methodical manner. The results are also supported with meticulous statistical analysis.

I recommend that the manuscript can be published after minor revision.

Response: We thank the Reviewer for finding the work novel and appropriately written, as well as the presented statistical analyses of high quality. We agree with both points raised and addressed them, accordingly, as detailed below with the respective changes in the manuscript **highlighted in green**:

Minor comments:

[R1-2] *Authors used polystyrene beads for the normalization of the cell density measurement (Fig. 1d). I suppose that these particles are not optimum objects to be used for this normalization, because both chemical composition and the shape of beads differ significantly from these parameters of bacteria. Ideally, live but non-dividing E. coli bacteria should be used for normalization. In that case, the neutral buffers such as phosphate-buffered saline (PBS) should be used instead of growth media. I think that authors should perform such a simple normalization experiment and add this data to the manuscript.*

Response: We agree and, first, would like to clarify that the cell data reported in **Fig. 1d** are normalized with respect to $t = 0$ (i.e., at the time of birth). Concomitantly, the polymer particle data in the same graph compare the density response of growing cells with the technical noise of the apparatus. In this context, several polymer particles were measured for three (3) hours, the period used for the single-cell measurements presented in this manuscript.

Beyond that, we agree with the Reviewer that a technical noise comparison with an object of similar composition and refractive index would be more appropriate. To this end, we first assessed the suggested use of the PBS neutral buffer. Here, we observed that a three-hour exposure to PBS yielded approximately 16% death rates, most likely due to the absence of nutrients. These death rates suggest that cellular composition likely changes during this period. These results are displayed below:

Increasing death rates of E. coli DH5 α as a function of exposure time to PBS. To perform this measurement, exponentially growing cells (37°C) in Mueller Hinton broth (Difco 275730, BD) were washed 3x in PBS and then transferred to PBS at the same dilution and temperature. Cell were sampled at the specified timepoints, stained with propidium iodide (~1 μ M), and transferred to an agarose pad for imaging (100x/1.4, Leica). It can be observed that an increasing % of death occurs, with each timepoint corresponding to approximately $n = 1,000$ observations.

Given the increased death rate of *E. coli* in PBS, we selected the imaging of fixed cells (2% glutaraldehyde) to perform the technical noise assessment. Our rationale here is that the composition of fixed cells is not expected to change in the three-hour window, while exhibiting a comparable refractive index to live cells. Using fixed cells, we observe that the technical noise is still substantially lower than the response of living cells. Accordingly, we revised **Fig. 1d** and its caption to reflect this change:

(d) Density, mass, and area growth curves of individual *E. coli* cells from birth to division; all parameters are normalized at $t = 0$ and color-coding represents the dynamics in size, mass, and density of the same cell (i.e., the curves represent the dynamics of individual cells and not several cells pooled together); horizontal red line denotes the dynamics of the normalized density (with respect to $t = 0$) of 10 fixed *E. coli* cells (DH5 α , fixed by incubation with 2% glutaraldehyde overnight), followed by 3x washing in PBS) over time (red line denotes the average and blue-shaded area denotes the 95% confidence intervals).

[R2-2] Some technical details are missing in the description (in the SI) of the microfluidic experiments. Authors forgot to mention what is the thickness of biopolymer microarray used for microfluidic 1D cultivation of bacteria, and how this thickness was determined during the casting of this microarray. Also, the diameters of the microfluidic inlet and outlet

should be provided. Finally, the description of how the layers of the microfluidic system were held together mechanically is not clear, i.e. whether clamps were used or what was the way to keep these parts together.

Response: We agree and performed the following revisions to address this comment:

Revision 1: Methods section

From (page 14, line 295): “The 1D microarrays were fabricated by electron beam lithography in SU8, subsequently transferred to PDMS, and then to a UV curable polymer that was index matched to water (Bio-133, My Polymers). Nutrients were provided by a doped membrane or a microfluidic channel.”

To (page 16, line 334): “The 1D microarrays were fabricated by electron beam lithography in SU8 (using our previously reported procedure⁷¹), subsequently transferred to PDMS, and then to a UV curable polymer that was index matched to water (Bio-133, My Polymers). The total thickness of the polymer film after imprinting was approximately 0.5 mm, thus, accommodating the working distance of all imaging objectives employed in this work. This thickness was determined by depositing approximately 300 μ L of the liquid prepolymer on the stamp. Nutrients were provided by a doped membrane or a microfluidic channel.”

Revision 2: Supplementary Information

From (page 4, line 63): “Our second method for nutrient/antibiotics delivery applied in the ampicillin experiments employed microfluidics assembled using five distinct parts that were held together mechanically^{4, 5} (**Sup. Fig. 1**): **(1)** a PDMS stamp (dimensions: 35 x 20 x 4 mm³, L x W x H) prepared by optical and cast molding lithography containing the microchannel (dimensions: 11 x 0.5 x 0.1 mm³, L x W x H); **(2)** the 1D microarray to immobilize cells; **(3)** a hybrid membrane (detailed below) to confine cells in the vertical direction; **(4)** two thin aluminum plates that hold parts (1), (2), (3), and the cell-loaded 1D

microarrays together; and **(5)** a syringe pump (Harvard Apparatus) connected to the microfluidics via Tygon tubing (1548XL, IDEX) set at 20 $\mu\text{l}/\text{min}$.”

To (page 4, line 66): “Our second method for nutrient/antibiotics delivery applied in the ampicillin experiments employed microfluidics assembled using five distinct parts that were held together mechanically^{4, 5} (**Sup. Fig. 1**): **(1)** a PDMS stamp (dimensions: 35 \times 20 \times 4 mm³, L \times W \times H) prepared by optical and cast molding lithography containing the microchannel (dimensions: 11 \times 0.5 \times 0.1 mm³, L \times W \times H); **(2)** the 1D microarray to immobilize cells; **(3)** a hybrid membrane (detailed below) to confine cells in the vertical direction; **(4)** *two thin aluminum plates that hold parts (1), (2), (3), and the cell-loaded 1D microarrays together using four screws symmetrically placed on the plates;* and **(5)** a syringe pump (Harvard Apparatus) connected to the microfluidics via Tygon tubing (1548XL, IDEX) *and 21 gauge needle tips,* set at 20 $\mu\text{l}/\text{min}$.”

Reviewer #3

I consider the manuscript by Nemati et al. to be of high quality and suitable for publication in Communication in Biology. The presentation and the format of the manuscript is appealing, while the scientific content is of high quality. The experimental and image/data analysis methods are smart and precisely described. The findings and conclusions are well supported by the data. What I particularly find interesting the results on growth differentiation and linking density fluctuations to fitness.

Response: We thank the Reviewer for commenting on the scientific content and quality of this manuscript, as well as the interest in the results of growth differentiation, density fluctuations, and fitness.

Reviewers' comments:

Reviewer #1 (Remarks to the Author):

The authors have made significant improvements in the paper and I trust that its scope is appropriate for this journal. The experimental technique is novel and useful. However, there are still major issues with some of the interpretation of the data that should be addressed before publication, as detailed below.

- The authors discuss "fitness levels of single-cells": this does not make sense, and is an ill-defined term. Fitness is defined at the population level, and also then the details of the growth protocols matter. For exponential population growth previous works have shown that the relevant properties at the single-cell level are not the doubling time but rather the growth rate (see for example Lin et al. *cell systems* 2017, Barber et al., *PLOS Computational Biology* 2021), though as far as I know previous works did not distinguish mass and size growth rates. Note that these works also showed that cellular noise decreases rather than increases the population fitness, in contrast to the interpretation of refs. 24-25. In light of these results, the discussion on line 201-212 should be revised. It seems to me that the main claim the paper makes in this context is to correlate single-cell doubling time with density fluctuations? If so, replacing "fitness" with "doubling time" will be both clearer and will avoid the above issues (though I couldn't follow the authors' logic on lines 243-245 as to the plausibility of observing such correlations).

- I have a hard time following the logic of the initial discussion of density homeostasis. The authors say that the fact that cells with $\gamma_m > \gamma_A$ have higher density at division than birth is evidence for a density homeostasis mechanism. But this seems to be an immediate consequence of the definition of γ_m and γ_A , unrelated to the mechanism of density homeostasis - which surely exists. Only around line 180 they discuss what I find to be compelling evidence for density homeostasis, namely, the negative correlation of $\gamma_m - \gamma_A$ with density.

- The authors should emphasize more that what they define as density is mass over area, as this is definitely an unconventional choice and would likely lead to misinterpretations.

- Under the assumption of exponential growth, eq. 17 is equivalent to eq. 11, and as such there is no way to distinguish the two models. Without noise (which the authors do not include) both of these models will lead to a constant density.

Minor

- The author discusses the possibility of density variability as potentially arising from asymmetric partitioning of biomolecules at division. This seems unlikely due to the high number of molecules partitioned, and would lead to strong negative correlations between sister cells which I don't think are observed experimentally?

Also, are density fluctuations significantly smaller upon division inhibition, as would be the expectation if this mechanism is playing a role?

- "single-cell analyses reveal considerable cell-to-cell variability in dry-density, as displayed in Fig. 1a with a coefficient of variation of 9%"

This is pretty much as low as it gets in single-cell studies, so different phrasing should be used.

- "Moreover, mass-based investigations have revealed that the growth-rate of mammalian cells is not constant across the cell cycle": note that this was also claimed for *B. subtilis* (Nordholt et al., *Current Biology* 2020) and *E. coli* (Kar et al., *bioRxiv* <https://doi.org/10.1101/2021.07.27.453901>).

Reviewer #2 (Remarks to the Author):

Authors performed additional experiments that I asked for, and they also addressed other minor remarks by adding explanation to the manuscript. I recommend publication of the current version.

Reviewer #1:

The authors have made significant improvements in the paper and I trust that its scope is appropriate for this journal. The experimental technique is novel and useful. However, there are still major issues with some of the interpretation of the data that should be addressed before publication, as detailed below.

General Response: We thank the Reviewer for deciding on the appropriateness of our manuscript for this journal. We would also like to acknowledge the Reviewer's additional efforts in this, 2nd round of revisions that identified concerns not identified in the 1st round. Below, we address all including the corresponding changes in the revised manuscript **highlighted in green**:

Main points:

[R1-1] *The authors discuss "fitness levels of single-cells": this does not make sense and is an ill-defined term. Fitness is defined at the population level and also then the details of the growth protocols matter. For exponential population growth previous works have shown that the relevant properties at the single-cell level are not the doubling time but rather the growth rate (see for example Lin et al. cell systems 2017, Barber et al., PLOS Computational Biology 2021), though as far as I know previous works did not distinguish mass and size growth rates. Note that these works also showed that cellular noise decreases rather than increases the population fitness, in contrast to the interpretation of refs. 24-25. In light of these results, the discussion on line 201-212 should be revised. It seems to me that the main claim the paper makes in this context is to correlate single-cell doubling time with density fluctuations? If so, replacing "fitness" with "doubling time" will be both clearer and will avoid the above issues.*

Response: The Reviewer suggests that the term "fitness of single-cells" is ill-defined and that it should be removed. Here, we need to clarify that others introduced this term (e.g., Science 359, 1283 and Proceedings of the National Academy of Sciences 108,

E67, cited as **REF 26** and **42** in the manuscript). Further, one other Reviewer found the link between density fluctuations and fitness interesting.

During the 1st round of revisions, we replaced the ‘*single-cell fitness*’ with “*reproduction rate*” in the Title, Abstract, and Results section, with the exception of a brief mention in the Discussion section. Since the scope of this manuscript is not to re-define an established term, but rather indicate that additional layers of regulation that may be at play, we performed the following revision, including the addition of the suggested computational reference as **REF 21**:

From (page 13, line 278): “Further, given the emerging relationship between reproduction rates of single-cells to the overall population fitness^{18, 19}, we note that the aforementioned notion of single-cell reproduction rates coincides with the Malthusian fitness equivalent of a single-cell⁴³. Our rationale behind this adaptation is that since Malthusian fitness is valid for a population, it should also be valid for its individual members. Overall, Malthusian fitness for a population is defined by its rate of growth (i.e., dN/dt , with N denoting the number of individuals of the population)⁴³. As such, the Malthusian fitness of a single-cell reduces to τ^{-1} , which denotes the inverse of the cycle duration or the reproduction rate of a single-cell, as also posited elsewhere⁶⁸. To distinguish the fitness of a single-cell from that of a population, we propose the term “instantaneous fitness effect”, with instantaneity denoting a moment or instant in phenotypic-space⁶⁹...”

To (page 13, line 278): “The explicit link between overall population fitness to the reproduction rates of single-cells^{18, 19, 21} suggest that the density fluctuations we observe here may also play a role to the fitness of a population. More explicitly, the Malthusian fitness of a population, defined by its rate of growth (i.e., dN/dt , with N being the number of individuals of the population)^{44, 69}, reduces to the rate of reproduction (τ^{-1}) for $N = 1$ ⁷⁰. Reducing Malthusian fitness to the single-cell level (i.e., $N = 1$) has been attempted before²⁷, approximated through the rate of elongation (γ_{size}) of single-cells; however, our results suggest that this approximation may not encompass all aspects of reproduction timing of single-cells.”

[R1-2] *I couldn't follow the authors' logic on lines 243-245 as to the plausibility of observing such correlations.*

Response: Following the revisions detailed in [R1-1], this section is no longer included.

[R1-3] *I have a hard time following the logic of the initial discussion of density homeostasis. The authors say that the fact that cells with $\gamma_m > \gamma_A$ have higher density at division than birth is evidence for a density homeostasis mechanism. But this seems to be an immediate consequence of the definition of γ_m and γ_A , unrelated to the mechanism of density homeostasis - which surely exists. Only around line 180 they discuss what I find to be compelling evidence for density homeostasis, namely, the negative correlation of $\gamma_m - \gamma_A$ with density.*

Response: We revised to more clearly state that the relationship between γ_A and γ_M only suggests density homeostasis, with the mechanism confirmed subsequently in the manuscript:

From (page 8, line 173): “We reasoned that this observation represents a density homeostasis mechanism, where density fluctuations maintain cellular density closer to the population average.”

To (page 9, line 176): “We reasoned that this observation *is potentially linked to a density homeostasis mechanism, where density fluctuations maintain cellular density closer to the population average.*”

[R1-4] *The authors should emphasize more that what they define as density is mass over area, as this is definitely an unconventional choice and would likely lead to misinterpretations.*

Response: The **Methods** section details that we define dry-density from the measured optical phase (**Eq. 1**), as reported by others (**REF 32**). We then quantify cell mass using the product of dry-density with area. This approach was first reported in 1952 (**REF 46**),

specifically for optical cell mass measurements. To improve the educational merit of this manuscript, we revised as:

Revision 1: Results section

From (page 6, line 118): “The combination of the “invisible” 1D microarray with spatial light interferometric imaging (SLIM) revealed that while both size (approximated by cell area, see **Methods**) and mass of single *E. coli* increased exponentially (**Fig. 1d**, inset and **Sup. Fig. 3**), cellular dry-density was not constant during growth (**Fig. 1d**).”

To (page 6, line 117): “The combination of the “invisible” 1D microarray with spatial light interferometric imaging (SLIM) enabled the precise quantification of cellular size (approximated by its area, see **Methods**), dry-density (directly from phase delay measurements³², **Methods**), and dry-mass (through the area product with density⁴⁶, **Methods**). This analysis revealed that while the size and mass of single *E. coli* cells increased exponentially (**Fig. 1d**, inset and **Sup. Fig. 3**), cellular dry-density was not constant during growth (**Fig. 1d**).”

Revision 2: Methods section

From (page 21, line 453): “We note that eq. 1 computes the area-based density of single-cell without essentially requiring prior knowledge of the cell area. This approach have been previously demonstrated in³¹ and⁴⁵.”

To (page 21, line 453): “We note that eq. 1 computes the density of single-cell without knowledge of the cell area, and applies 2D maximum projection of 3D data (planes P_{i-1} – to P_{i+1} , where i is the best focus plane – see *Image Analysis*). Finally, to determine cell mass, we multiplied cell density (Eq. 1) with the cell area. This has been previously demonstrated in cell mass measurements, where cell thickness cannot be determined as accurately its area, given the lower axial than planar resolution of most optical systems⁴⁶.”

[R1-5] *Under the assumption of exponential growth, eq. 17 is equivalent to eq. 11, and as such there is no way to distinguish the two models. Without noise (which the authors do not include) both of these models will lead to a constant density.*

Response: We agree that both models lead to similar differential equations, but they have different proportionality factors and, thus, each leads to a distinct prediction. Specifically, the right-hand side of Eq. 11 is $\alpha \cdot M$, and predicts increasing density with γ_{size} . In contrast, the right-hand side of Eq. 17 is $\alpha \cdot [dM/dt]$, and predicts increasing density with the level of differentiation, $\gamma_{\text{size}}/\gamma_{\text{mass}}$. Our empirical data show a stronger increase of density with γ_{size} (**Sup. Fig. 7b**) rather than $\gamma_{\text{size}}/\gamma_{\text{mass}}$ (**Sup. Fig. 7a**).

Minor

[R1-6] *The author discusses the possibility of density variability as potentially arising from asymmetric partitioning of biomolecules at division. This seems unlikely due to the high number of molecules partitioned, and would lead to strong negative correlations between sister cells which I don't think are observed experimentally? Also, are density fluctuations significantly smaller upon division inhibition, as would be the expectation if this mechanism is playing a role?*

Response: The Reviewer suggests that:

- i. Asymmetric partitioning of biomolecules is an unlikely driver of density variability.
- ii. Density differences between sisters to their mother should be negatively correlated.
- iii. Density fluctuations should be smaller upon division inhibition.

Here, we clarify that:

- i. As detailed in the original and revised manuscripts, the hypothesis of asymmetric partitioning of biomolecules is based on observations by others (**REF 58**) that focus on multiple copies of a single gene product.
- ii. Density differences between sisters with their mother are uncorrelated ($\rho_{\text{Pearson}} = 0.12$, $p = 0.11$), possibly due to the innate size variability. Mass differences, a

more direct relevant measure, are indeed uncorrelated ($\rho_{\text{Pearson}} = -0.68$, $p \ll 0.001$).

- iii. Both the original and revised manuscripts detail one of our central findings that density fluctuations subside upon division inhibition (please refer to **Fig. 3a** and **Sup. Fig. 9**).

We revised to add a graph as an inset to **Fig. 2d** depicting the negative correlation of the mass differences between daughters with their mother:

Fig. 2d: ...Inset plots the daughter-daughter correlation of the mass differences (d_1 and d_2 , in %) with their mother, with each replicate coded with a distinct color.

[R1-8] "single-cell analyses reveal considerable cell-to-cell variability in dry-density, as displayed in Fig. 1a with a coefficient of variation of 9%" This is pretty much as low as it gets in single-cell studies, so different phrasing should be used.

Response: We replaced "considerable" with "non-negligible".

[R1-9] "Moreover, mass-based investigations have revealed that the growth-rate of mammalian cells is not constant across the cell cycle": note that this was also claimed for *B. subtilis* (Nordholt et al., *Current Biology* 2020) and *E. coli* (Kar et al., *bioRxiv* <https://doi.org/10.1101/2021.07.27.453901>).

Response: We acknowledge the suggestion of these two references, which, however, do not perform quantitative mass measurements of single-cells. As such, we did not include them given the sentence in question pertains to “*mass-based investigations*”.

Reviewer #2:

Authors performed additional experiments that I asked for, and they also addressed other minor remarks by adding explanation to the manuscript. I recommend publication of the current version.

Response: We thank the Reviewer for originally commenting on the scientific content and quality of this manuscript, and, now, for recommending publication.